# Multi-Location-Aware Joint Optimization of Content Caching and Delivery for Backhaul-Constrained UDN

**DOI:** 10.3390/s19112449

**Published:** 2019-05-29

**Authors:** Wenpeng Jing, Xiangming Wen, Zhaoming Lu, Haijun Zhang

**Affiliations:** 1Beijing Key Laboratory of Network System Architecture and Convergence, Beijing Laboratory of Advanced Information Networks, Beijing University of Posts and Telecommunications, Beijing 100876, China; xiangmw@bupt.edu.cn (X.W.); lzy_0372@163.com (Z.L.); 2Beijing Advanced Innovation Center for Materials Genome Engineering, Beijing Engineering and Technology Research Center for Convergence Networks and Ubiquitous Services, School of Computer and Communication Engineering, University of Science and Technology Beijing(USTB), Beijing 100083, China; haijunzhang@ieee.org

**Keywords:** mobile edge caching, content placement, user association, resources allocation

## Abstract

Mobile edge caching is regarded as a promising way to reduce the backhaul load of the base stations (BSs). However, the capacity of BSs’ cache tends to be small, while mobile users’ content preferences are diverse. Furthermore, both the locations of users and user-BS association are uncertain in wireless networks. All of these pose great challenges on the content caching and content delivery. This paper studies the joint optimization of the content placement and content delivery schemes in the cache-enabled ultra-dense small-cell network (UDN) with constrained-backhaul link. Considering the differences in decision time-scales, the content placement and content delivery are investigated separately, but their interplay is taken into consideration. Firstly, a content placement problem is formulated, where the uncertainty of user-BS association is considered. Specifically, different from the existing works, the specific multi-location request pattern is considered that users tend to send content requests from more than one but limited locations during one day. Secondly, a user-BS association and wireless resources allocation problem is formulated, with the objective of maximizing users’ data rates under the backhaul bandwidth constraint. Due to the non-convex nature of these two problems, the problem transformation and variables relaxation are adopted, which convert the original problems into more tractable forms. Then, based on the convex optimization methods, a content placement algorithm, and a cache-aware user association and resources allocation algorithm are proposed, respectively. Finally, simulation results are given, which validate that the proposed algorithms have obvious performance advantages in terms of the network utility, the hit ratio of the cache, and the quality of service guarantee, and are suitable for the cache-enabled UDN with constrained-backhaul link.

## 1. Introduction

The mobile data traffic has increased dramatically in the recent decades, which puts a great strain on the mobile network [1]. To meet the ever-increasing traffic demand, the small-cell, especially ultra dense network (UDN) is regarded as one of the most effective ways to boost the mobile network throughput [2,3]. Considering the huge number of small-cell base stations (BSs) in UDN, to equip each small-cell base station (SBS) with a high capacity backhaul link, such as optical fiber, would be a huge monetary cost for mobile network operator [4]. Furthermore, with the advance of wireless transmission technologies such as massive Multiple-Input Multiple-Output (MIMO) and millimeter wave (mmWave) communication, the backhaul link instead of the wireless link, is more likely to become the bottleneck of the data transmission link [5]. Hence, how to reap the benefits of UDN without the constraint of high-capacity backhaul link is a critical problem.

Notably, the explosive growth of mobile traffic mainly derives from the ever-increasing demand for mobile video [1]. Meanwhile, a large portion of the mobile traffic is contributed by the duplicate transmission of the same contents, especially the popular video contents. Mobile edge caching, which means that the BS is equipped with a storage device cache of the popular content and serves users locally without backhaul transmission [6,7], has great potential to reduce the backhaul traffic burden. Due to the relatively low cost of installing storage device, mobile edge caching would be a cost-effective substitute solution to the high-speed backhaul link. It is anticipated that the mobile edge caching would be able to save up to 35% on backhaul usage for the mobile network, and reduce the latency in mobile services by 50% [4]. Furthermore, considering that the mobile traffic of each BS always has temporal fluctuation property, the BSs can prefetch the video contents during the off-peak hours, and provide simultaneous video transmission with low backhaul requirements during the peak hours. As a result, to equip the SBSs with caching capability has great benefits for the UDN with capacity-constrained backhaul link.

Generally, the introduction of mobile edge caching would lead the video content transmission to two phases: the content placement phase and the content delivery phase. The content placement phase usually occurs in off-peak hours. Utilizing the light-load backhaul link, the SBSs would proactively fetch the contents based on users’ demand. The content delivery phase occurs when users send content requests. If the content requested has been cached (which is called as cache hit), it would be delivered to the user from the SBS’s cache directly without the backhaul transmission. However, if the content has not been cached (which is called as cache miss), it would be fetched from the remote server through the backhaul link and then delivered to the user.

Considering the characteristics of the cache-enabled UDN with constrained-backhaul, it is not easy to fully reap the mobile edge caching benefits. Specifically,

1The cache capacity of each SBS in UDN is limited, which means that the traditional content placement scheme proposed for wired networks (e.g., content distribution network (CDN) [8]) can not be applied directly. Considering the great volume of the video contents, it becomes more critical for the caching strategy to achieve high cache hit ratio in cache-enabled UDN.2The coverage of each SBS in UDN is likely to be small and overlapping with each other. This means that there are likely multiple SBSs that the user can be associated with. For cache-enabled UDN, the user-SBS association decision can be made based on not only the signal strength and the load of the SBS, but also the local availability of the content requested by users. But the flexibility of user-SBS association also introduces uncertainty for predicting the content demand of each SBS.3The content popularity reflects the overall preference of a large number of users. At the same time, each user has its own specific preference. Considering each SBS in UDN only serves a limited number of users, and the aggregation of such a few users’ demand is not always consistent with the global popularity [9], the content popularity-based caching schemes can not be applied to the cache-enabled UDN scenario directly.4The users in UDN are mobile, and can send content requests from any locations in the network. This gives further uncertainty in characterizing the probability of certain content to be requested in each SBS.

In general, the limited cache capacity of each SBS, the uncertainty of users’ location, the diverse content preference of each user, and the flexible user-SBS association relationship, make the content placement and content delivery become a challenging problem remaining to be solved.

### 1.1. Related Work

Caching techniques have been studied extensively in wired CDN scenarios [8,10]. However, different from the BSs’ cache in wireless networks, the CDN always has large-sized cache and can store millions of videos. Moreover, the network topology of CDN is fixed and the transmission link is stable, which has discrepancies with the wireless networks. As a result, the caching schemes proposed for CDN are not suitable for the cache-enabled wireless networks. On the other hand, the user-BS association is a topic that has been studied by a lot of papers [11]. Instead of Max-SINR-based schemes, the load-balance-based user-BS association scheme is investigated for wireless networks, especially the heterogeneous networks (HetNet), and can improve the network performance significantly [12,13,14]. However, most of these existing works do not consider the local availability of the contents enabled by the mobile edge caching, which is also not suitable for the cache-enabled wireless networks.

Considering the interplay of caching and user-BS association, a few papers investigate the optimization of the caching strategy and user association schemes jointly. The pioneering paper [15] validates the effectiveness of the mobile edge caching, and demonstrates that up to 500% of the system throughput gain can be achieved by the joint optimization of caching and user association scheme. In [16], based on a weighted sum utility of the data rates and backhaul traffic reduction, a content placement and user association algorithm is proposed, which could achieve load balancing and backhaul saving. In [17], the total time to satisfy users’ demand is minimized by joint optimization of caching and interference coordination in a cache-enabled HetNet with wireless backhaul. In [18], the traffic served by Macro BSs (MBSs) is minimized by the user association and caching optimization. In [19], the transmission time required to satisfy all users’ demand is minimized by joint optimization of caching, routing, and channel assignment. In [20], the caching and user association policy is optimized to minimize the average download delay, where both the wireless link delay and the backhaul link delay are considered.

The papers mentioned above investigate the content caching and user-BS association schemes, based on the objective of minimizing user-perceived delay, MBSs’ traffic load, or maximizing user data rates and BSs’ backhaul traffic reduction. However, there are two aspects that have been neglected by all of them. (1) Specifically, all the papers assume that users would request contents from one fixed location in the network, which is not in line with the practical request pattern of mobile users. Indeed, the statistical results in [21,22] show that most of the users periodically initiate content requests from certain set of cells, rather than from only one specific location. If users’ multi-location request pattern is not considered in the caching scheme optimization, the network performance would be deteriorated. (2) In addtion, most of these works do not account for the backhaul bandwidth constraints. However, the users of cache wrongly associated with the same SBS would have to share the limited backhaul bandwidth of the SBS to transmit their contents. Without proper allocation of the wireless and backhaul resources, there is a high probability that network performance would be constrained by the backhaul link during the content delivery phase. Consequently, the proposed schemes without backhaul constraint cannot be applied to the backhaul-constrained UDN scenario.

The papers [21,23] integrate the multi-location request pattern into the problem formulation and design multi-location-aware content caching schemes, respectively. However, paper [21] adopts a Max-SINR-based user-BS association, and paper [23] assumes that each user would be associated with the nearest SBS that has cached the requested content. This kind of fixed user-BS association scheme reduces the flexibility and limits the performance gain of caching. Hence, it is still a challenging problem that how to optimize the content caching and delivery schemes by considering the user-association, users’ location uncertainty, as well as the wireless and backhaul resources constraints jointly.

### 1.2. Contribution and Organization of the Paper

In this paper, we investigate the optimization of content placement and content delivery schemes in cache-enabled UDN with constrained-backhaul. Due to the differences in decision time-scales, the content placement and delivery problem is formulated and investigated separately, but their interaction is integrated into each other. Specifically, during the content placement phase, the content placement is optimized based on the probability of user’s location and the probability of user requesting specific content. During the content delivery phase, given the fixed content cache status, the user-SBS association and resources allocation are optimized. The major contributions of this paper are summarized as follows:1A content placement problem is formulated, which aims to optimize the hit ratio of each SBS’s cache. Different from the existing works which assume users would request the video from one specific location, this paper considers the more realistic multi-location request pattern of mobile users. Taken into account the uncertainty of user-SBS association, the content placement problem is optimized in conjunction with the user-SBSs association probability. Considering the mix-integer nature of the problem, a content placement solution is derived based on the iterative scheme.2A cache-aware joint user association and resources allocation optimization problem is formulated, of which the objective is to maximize the sum of logarithmic each user’s data rates. Considering the NP-hard nature, a variable relaxation technique is adopted to convert the original problem into a convex optimization problem. Based on Lagrangian dual method, an optimal user-association and resources allocation solution is derived, and its property is also analyzed.

The remainder of the paper is organized as follows: Section 2 presents the system model, and the content placement and user-SBS association optimization problem are formulated, respectively. Section 3 proposes the content placement algorithm and cache-aware user association and resources allocation algorithm, respectively. In Section 4, the simulation results are presented to evaluate the performance of the proposed algorithms. Finally, Section 5 gives the conclusions of this paper.

## 2. System Model

The downlink transmission of a cache-enabled UDN scenario is considered, where *M* SBSs are densely deployed to serve *K* users. The set of users and BSs are denoted as $K=\left\{1,2,\cdots ,K\right\}$ and $M=\left\{1,2,\cdots ,M\right\}$, respectively. Assume that both the SBSs and users are equipped with only one antenna. Meanwhile, the backhaul links of the SBSs are bandwidth-limited, but all the SBSs are equipped with the mobile edge caching devices. The system bandwidth is BW, which is divided into *N* resource blocks (RBs). The system bandwidth is reused by all the SBSs in the UDN.

### 2.1. User Behavior of Requesting Contents

Different from most of the existing works, in this paper the multi-location request pattern of users is considered, which means that each user is likely to send the requests from any locations of the network. Specifically, the probability of user k∈K issuing requests from SBS m∈M is denoted as ${\mathrm{Pr}}_{m}^{k}$. Without loss of generality, the overall possibilities of user *k* sending requests over all the small-cells can be denoted as ${\mathrm{Pr}}^{k}=\left\{{\mathrm{Pr}}_{1}^{k},\cdots ,{\mathrm{Pr}}_{m}^{k},\cdots ,{\mathrm{Pr}}_{M}^{k}\right\}$. Note that the possibilities ${\mathrm{Pr}}^{k}$ can be predicted based on each user’s request logs by the operators or the video service providers. Combined with the techniques of human movement predication or point of interest (PoI) recommendation system [24,25], the fine-grained location information where a mobile user sends the requests can even be predicted. As a result, this paper assumes that each user would send the requests from multiple small-cells, and there would be one specific location for each small-cell that the user would send the requests. Specifically, the ${\mathrm{Pr}}_{m}^{k}$ can also be used to denote the probability that user *k* would send the content requests from the specific location in the coverage of SBS *m*.

Assume that the overall contents library consists of *F* contents, which can be denoted as set F={1,2,⋯,F}. For ease of illustration, all contents are assumed to have the same size. The capacity of each SBS’s cache is denoted as $C^{MAX}$, which means that at most $C^{MAX}$ contents can be stored. If a content requested by the user has been cached by the associated SBS, the content would be delivered to the user directly from the SBS’s cache. Otherwise, the content would be fetched from the remote content server via the backhaul link.

Denote $p^{k,f}\in \left[0,1\right]$ as the probability of user *k* sending requests for content f∈F. It should be noted that the value of $p^{k,f}$ reflects the user’s individual preference for contents, which is not the same with the content popularity. $\left\{p^{k,f}\right\}$ could be predicted based on machine learning techniques [26], and is assumed to be known a priori during the content placement phase.

### 2.2. Wireless Transmission Model

If user *k* located in the coverage of SBS n∈M is associated with SBS *m* and allocated with one RB, the maximum data rates of user *k* can be calculated based on Shannon capacity, i.e.,
(1)$$R_{m,n}^{k}=B\log_{2}\left(1+\frac{p_{m}h_{m,n}^{k}}{\sigma^{2}+I_{m,n}^{k}}\right),$$
where *B* is the bandwidth of each RB, $p_{m}$ is the transmit power of SBS *m*, $h_{m,n}^{k}$ is the channel power gain from SBS *m* to user *k* located in the coverage of SBS *n*, $\sigma^{2}$ is the power of the additive white Gaussian noise, $I_{m,n}^{k}=\sum_{m^{\prime }\in M_{I}^{k}}p_{m^{\prime }}h_{m^{\prime },n}^{k}$ is the power of inter-cell interference suffered by user *k*, and $M_{I}^{k}$ represents the set of SBSs that would generate interference to user *k*.

## 3. Problem Formulation

The overall goal of this paper is to design content caching and delivery schemes that can take full advantages of the limited storage resources (i.e., the cache) and communication resources (including the wireless links and the backhaul links) to improve users’ data rates. However, considering the differences in the decision time-scales, the corresponding content placement optimization problem and content delivery optimization problem are investigated separately and respectively.

### 3.1. Content Placement Problem

For backhaul-constrained UDN, there is a high probability for the backhaul link to be the bottleneck of the transmission link, especially during the peak traffic hour. Thus, how to improve the percentage of the traffic offloaded by the cache, i.e., to maximize the hit ratio of the cache, is the main objective for the content placement scheme.

Denote $x_{m}^{f}\in \left\{0,1\right\}$ as an indicator variable, which denotes whether the content *f* is cached by SBS *m* or not. Then, taking into consideration the uncertainty of users’ location and each user’s individual preference on the content, the hit ratio of SBS *m* can be calculated as
(2)$$U_{m}=\sum_{k\in K}\sum_{n\in M}{\tilde{a}}_{m,n}^{k}{\mathrm{Pr}}_{n}^{k}\sum_{f\in F}p^{k,f}x_{m}^{f},$$
where the ${\tilde{a}}_{m,n}^{k}$ is the auxiliary variable indicating the association probability that user *k* located in the coverage of SBS *n* would be associated with SBS *m*. Even though the user *k* is located in the coverage of SBS *n*, it can be associated with any one of the SBSs.

Then, the content placement problem is formulated as
(3)$$\begin{array}{cc} P1 & :\max_{\left\{x_{m}^{f},{\tilde{a}}_{m,n}^{k}\right\}}\sum_{m\in M}\log U_{m}\\ s.t. & :\\ C1 & :\sum_{f\in F}x_{m}^{f}=C^{Max},\forall m\in M,\;\\ C2 & :x_{m}^{f}\in \left\{0,1\right\},\forall m\in M,f\in F,\\ C3 & :\sum_{m\in M}{\tilde{a}}_{m,n}^{k}=1,\forall k\in K,n\in M\\ C4 & :{\tilde{a}}_{m,n}^{k}\in \left[0,1\right],\forall k\in K,n\in M,m\in M,\\ C5 & :\sum_{k\in K}\sum_{n\in M}\frac{{\mathrm{Pr}}_{n}^{k}{\tilde{a}}_{m,n}^{k}R^{k,Th}}{R_{m,n}^{k}}=N,\forall m\in M. \end{array}$$
The sum of logarithmic hit ratio of each SBS is adopted as the objective function in problem P1. The improvement of the hit ratio means more backhaul traffic reduction, which would contribute to the improvement of users’ data rates, especially for the backhaul-constrained UDN scenarios. Furthermore, the maximization of the sum of logarithmic hit ratio would give a proportional fairness guarantee for each SBS in terms of hit ratio. To some extent, more fair hit ratio means more fair backhaul reduction among SBSs, which results in more balanced backhaul traffic during the content delivery phase. The constraints C3 and C4 mean that there is not a unique BS association constraint. This is because that ${\tilde{a}}_{m,n}^{k}$ only denotes the probability of user-SBS association, and is not the realistic user association solution for the content delivery phase. The $R^{k,Th}$ in constraint C5 denotes each user’s minimum data rates requirement, and the constraint C5 means that the sum of the expected RB cost to satisfy all the associated users’ quality of service (QoS) requirements should not be larger than the overall available RBs. In other words, the constraint C5 ensures that the wireless link would not be the bottleneck during the content delivery phase.

It should be noted that ${\tilde{a}}_{m,n}^{k}$ is the auxiliary variable for obtaining the optimal content placement solution. Based on $\left\{{\tilde{a}}_{m,n}^{k}\right\}$, the interplay of content caching and the content delivery is integrated into the content placement problem.

### 3.2. Content Delivery Problem

After the content placement phase, the caching states of the contents, i.e., $\left\{x_{m}^{f}\right\}$ would be determined. It is assumed that the content delivery would operate in a time-slot manner and would be optimized slot by slot in this paper. In one time slot, each user would send one content request from one specific location.Because the locations from which the users send the requests would be fixed in one time slot, the *n* in the subscript of $R_{m,n}^{k}$ can be omitted without ambiguity. As users’ data rates would be impacted by the traffic load of SBSs’ wireless link and the backhaul link, the user-SBS association and wireless/backhaul resources should be considered jointly to provide high data rates for users.

Denote $a_{m}^{k}$ as the user-SBS association variable, which indicates whether user *k* is associated with SBS *m* or not. Meanwhile, denote $b_{m}^{k}$ as the resources allocation variable denoting how many RBs are allocated to user *k* by SBS *m*. With the objective of improving each user’s data rates during content delivery, a user-SBS association and resources allocation problem is formulated as
(4)$$\begin{array}{cc} P2 & :\max_{\left\{a_{m}^{k},b_{m}^{k}\right\}}\sum_{k\in K}\log\sum_{m\in M}a_{m}^{k}b_{m}^{k}R_{m}^{k}\\ s.t.:\\ C6 & :\sum_{m\in M}a_{m}^{k}\leq 1,\forall k\in K,\\ C7 & :a_{m}^{k}\in \left\{0,1\right\},\forall m\in M,k\in K,\\ C8 & :b_{m}^{k}\in \left\{0,\cdots ,N\right\},\forall m\in M,k\in K,\\ C9 & :\sum_{k\in K}b_{m}^{k}\leq N,\forall m\in M\\ C10 & :\sum_{k\in K}c_{m}^{k}a_{m}^{k}b_{m}^{k}R_{m}^{k}\leq B_{m}^{BH},\forall m\in M, \end{array}$$
where $B_{m}^{BH}$ is the backhaul bandwidth of SBS *m*, and $c_{m}^{k}\in \left\{0,1\right\}$ is a parameter indicating whether user *k* would cost the backhaul link if it is associated with SBS *m*. Specifically, if the content requested by user *k* has been cached by SBS *m*, then $c_{m}^{k}=0$. Otherwise, $c_{m}^{k}=1$. It should be noted that the $c_{m}^{k}$ would be a known parameter in each content delivery slot. The constraints C6 and C7 are the unique association limit, which means that each user must associate with only one SBS. The constraint C8 and C9 guarantee that the total number of RBs allocated by each SBS to all the associated users should not exceed the maximum number of available RBs. The constraint C10 is the backhaul bandwidth limit, ensuring that each SBS’s backhaul traffic should not be larger than its backhaul capacity.

The sum of the logarithmic user’s data rates is used as the objective function of P2, in order to provide proportional fairness among users. Moreover, a deterministic QoS guarantee for each user is not adopted in the problem P2. This is because the backhaul link is very likely to become the bottleneck of the transmission in backhaul-constrained UDN. In that case, the deterministic QoS guarantee for all the users cannot be achieved, and the problem with QoS constraint would have no solutions.

## 4. Solution to the Optimization Problem

Both the problem P1 and P2 have integer variables and are non-convex optimization problems, which are generally NP-hard and make it difficult to obtain the optimal solutions. As a result, we would convert the original problems into more tractable forms and design low-complexity algorithms, as described in the following.

### 4.1. Iterative Content Placement Algorithm

In order to solve the mixed-integer optimization problem P1, an iterative scheme is adopted that optimizes content placement and the user-SBS association probabilities iteratively and separately. Specifically, by fixing the user-SBS association probabilities, the content placement would be optimized. Then, on the condition that the content placement is given, the user association probabilities would be optimized. The process would be implemented iteratively until no performance improvement can be achieved.

On the condition that the content placement solution, i.e., $\left\{x_{m}^{f}\right\}$, is given, the problem P1 would be transformed into the following form
(5)$$\begin{array}{cc} P1.1 & :\max_{\left\{{\tilde{a}}_{m,n}^{k}\right\}}\sum_{m\in M}\log U_{m}\\ s.t. & :\\ C3\;\; & :\sum_{m\in M}{\tilde{a}}_{m,n}^{k}\leq 1,\forall k\in K,n\in M\\ C4 & :{\tilde{a}}_{m,n}^{k}\in \left[0,1\right],\forall k\in K,n\in M,m\in M,\\ C5 & :\sum_{k\in K}\sum_{n\in M}\frac{{\mathrm{Pr}}_{n}^{k}{\tilde{a}}_{m,n}^{k}R^{k,Th}}{R_{m,n}^{k}}\leq N,\forall m\in M, \end{array}$$
It can be easily derived that the objective function of problem P1.1 is convex with respect to the variable $\left\{{\tilde{a}}_{m,n}^{k}\right\}$. Meanwhile, the feasible set defined by constraints C3-C5 is also convex. Thus, problem P1.1 is a convex optimization problem, and the optimal solution, i.e., $\left\{{\tilde{a}}_{m,n}^{k*}\right\}$ can be obtained in polynomial time based on convex optimization methods such as interior point method or ellipsoid method [27].

Fixing the user-SBS association probability variables $\left\{{\tilde{a}}_{m,n}^{k}\right\}$, problem P1 would be reduced to the following form
(6)$$\begin{array}{cc} P1.2 & :\max_{\left\{x_{m}^{f}\right\}}\sum_{m\in M}\log U_{m}\\ s.t. & :\\ C1 & :\sum_{f\in F}x_{m}^{f}=C^{Max},\forall m\in M,\;\\ C2 & :x_{m}^{f}\in \left\{0,1\right\},\forall m\in M,f\in F. \end{array}$$
Obviously the utility of each BS is not coupled with each other when the user-SBS association probabilities are fixed. Consequently, to solve the problem P1.2 is equivalent to solve the following sub-problem for each SBS
(7)$$\begin{array}{cc} P1.3 & :\max_{\left\{x_{m}^{f}\right\}}U_{m}\\ s.t. & :\\ C1 & :\sum_{f\in F}x_{m}^{f}=C^{Max},\\ C2 & :x_{m}^{f}\in \left\{0,1\right\},\forall f\in F. \end{array}$$
The optimal solution can be easily derived as
(8)$$x_{m}^{f^{*}}=\left\{\begin{array}{c} 1,\mathrm{for}\;\;\;f^{*}=\arg\max_{f}\sum_{k\in K}\sum_{n\in M}{\mathrm{Pr}}_{n}^{k}p^{k,f}{\tilde{a}}_{m,n}^{k},\\ 0,\mathrm{else}. \end{array}\right.$$
The optimal solution is straightforward, and means that each SBS should cache the contents that can achieve the highest hit ratio.

Based on the derivation above, the overall content placement algorithm is designed, which is shown in the following Algorithm 1. The Algorithm 1 is an iterative algorithm, which includes two key steps, i.e., the step of calculating the ${\tilde{a}}_{m,n}^{k*}$ and the step of calculating the $x_{m}^{f*}$. For each loop, if the optimal ${\tilde{a}}_{m,n}^{k*}$ is obtained based on the interior point method, the corresponding computation complexity can be given by OMKlogMKMKt0εt0ε, where $t^{\left(0\right)}$ is a parameter of interior point method and ε is the tolerance of the solution. As for the step of calculating the $x_{m}^{f*}$, its complexity can be given by $O\left(MF^{2}\right)$. Let us denote $I_{\max}$ as the maximum number of iterations of the loop in Algorithm 1. We can give the overall computation complexity of Algorithm 1 by OImaxMKlogMKMKt0εt0ε+MF2. Besides, Algorithm 1 is a centralized algorithm, where all the procedures are implemented in a central entity, e.g., the remote server of the content provider. Specifically, the information of ${\mathrm{Pr}}_{n}^{k},p^{k,f}$, and $R_{m,n}^{k}$ is required to calculate the solution of P1. Note that similar with the traditional CDN, when the user sends request for certain content, the request would be sent to the server of the content provider firstly. Then, the content provider would decide to response the request by the remote server or by the cache of SBS the user associated with. Hence, the content provider would know all the information of the request, including user’s ID, the requested content ID, the time and the location of the user sending the request. Furthermore, all this information is known by the content provider in a real-time manner. Moreover, as long as the average data rates of the wireless link could be uploaded in a real-time manner by SBSs, the information based on which the content provider made the content placement decision, would be comprehensive and up-to-date. Consequently, Algorithm 1 could avoid the duplicate cache problem to some extent.

**Algorithm 1** Iterative Content Placement Algorithm**Initialize**xmf based on the global popularity, that is each SBS caches the most popular contents of the overall network;**repeat**
    Obtain the a˜m,nk* based on interior point method or ellipsoid method;     Obtain xmf* based on Equation (8);     T=T+1;**until** No performance improvement could be obtained or T≥Tmax.


### 4.2. Cache-Aware User Association and Resources Allocation Algorithm

The problem P2 is an integer programming problem, which is computationally prohibitive to find the optimal solution. To make the problem more tractable, the following transformation is introduced.

Firstly, the $\left\{b_{m}^{k}\right\}$ is relaxed to be the continuous variables ${\tilde{b}}_{m}^{k}\in \left[0,N\right]$, which means that any percentage of the radio resources can be allocated to one user. Then, the variable $d_{m}^{k}=\frac{a_{m}^{k}{\tilde{b}}_{m}^{k}}{N}$ is introduced, of which the range can be derived as $d_{m}^{k}\in \left[0,1\right]$. Based on $d_{m}^{k}$, the problem P2 can be converted into the the following form
(9)$$\begin{array}{cc} P3 & :\max_{\left\{d_{m}^{k}\right\}}\sum_{k\in K}\log\sum_{m\in M}d_{m}^{k}NR_{m}^{k}\\ s.t. & :\\ C3.1 & :\sum_{k\in K}d_{m}^{k}\leq 1,\forall m\in M,\\ C3.2 & :\sum_{m\in M}d_{m}^{k}\leq 1,\forall k\in K,\\ C3.3 & :d_{m}^{k}\in \left[0,1\right],\forall m\in M,k\in K,\\ C3.4 & :d_{m}^{k}d_{m^{\prime }}^{k}=0,\forall m\in M,m^{\prime }\in M,k\in K,\\ C3.5 & :\sum_{k\in K}c_{m}^{k}d_{m}^{k}NR_{m}^{k}\leq B_{m}^{BH},\forall m\in M. \end{array}$$
The constraints C3.2, C3.3, and C3.4 specify that each user could be associated with at most one SBS and be allocated with any fraction of the overall radio frequency resources. The constraints C3.1 and C3.5 specify that the allocated wireless frequency bandwidth and backhaul bandwidth cannot be larger than the overall available resources.

Notably, by replacing the variables $\left\{a_{m}^{k}\right\}$ and $\left\{b_{m}^{k}\right\}$ with $\left\{d_{m}^{k}\right\}$, the objective function of problem P2 is converted to be a convex function. Meanwhile, the feasible set defined by C3.1–C3.5 is also a convex set. These mean that the problem P3 is a convex optimization problem.

Next the Lagrangian dual method is adopted to solve the problem P3. Specifically, a closed-form optimal solution can be derived that is helpful to investigate the characteristics of the optimal user association and resources allocation. The Lagrangian function can be denoted as
(10)$$L\left(\left\{d_{m}^{k}\right\},\lambda ,\beta \right)=\sum_{k\in K}\log\sum_{m\in M}d_{m}^{k}NR_{m}^{k}+\sum_{m\in M}\lambda_{m}\left(1-\sum_{k\in K}d_{m}^{k}\right)+\sum_{m\in M}\beta_{m}\left(B_{m}^{BH}-\sum_{k\in K}c_{m}^{k}d_{m}^{k}NR_{m}^{k}\right)$$
where λ and β are non-negative Lagrangian multipliers for constraints C3.1 and C3.5, respectively. Then, the Lagrangian dual function can be expressed as
(11)$$\begin{array}{c} g\left(\lambda ,\beta \right)=\max_{\left\{d_{m}^{k}\right\}}L\left(\left\{d_{m}^{k}\right\},\lambda ,\beta \right)\\ s.t.:C3.2,C3.3,C3.4. \end{array}$$
and the Lagrange dual problem is
(12)$$\min_{\lambda ≻0,\beta ≻0}g\left(\lambda ,\beta \right)$$

According to the Karush–Kuhn–Tucker (KKT) conditions, the optimal solution of the problem P3 should satisfy
(13)$$\frac{\partial L\left(\left\{d_{m}^{k}\right\},\lambda ,\beta \right)}{d_{m}^{k}}=0.$$
Based on Equation (Equation 13), the optimal solution for user *k* can be derived as
(14)$$d_{m*}^{k}=\frac{1}{\lambda_{m*}+\beta_{m*}NR_{m*}^{k}c_{m*}^{k}}$$
for
(15)$$m*=\arg\max_{m^{\prime }}\log\frac{NR_{m^{\prime }}^{k}}{\lambda_{m}+\beta_{m^{\prime }}NR_{m^{\prime }}^{k}c_{m^{\prime }}^{k}},$$
Otherwise, $d_{m}^{k}=0$.

The Lagrangian multipliers can be updated using the ellipsoid method or subgradient method until convergence. Then the optimal solution of problem P3, i.e., $\left\{d_{m}^{k*}\right\}$ can be obtained easily. At last, the solutions corresponding to the original problem P2 could be obtained based on the following equations
(16)$$a_{m}^{k*}=\left\{\begin{array}{c} 1,\mathrm{if}\;d_{m}^{k*}>1,\\ 0,\mathrm{else}. \end{array}\right.$$
and
(17)$$b_{m}^{k*}=\left\{\begin{array}{c} N\cdot \mathrm{Round}\left(a_{m}^{k*}\right),\mathrm{if}\;d_{m}^{k*}>1,\\ 0,\mathrm{else}, \end{array}\right.$$
where $\mathrm{Round}\left(x\right)$ denotes the function that returns the nearest integer of *x*.

Based on the derivation, the cache-aware user association and resources allocation algorithm could be proposed, which is shown in the following Algorithm 2. Because of the variables’ relaxation and the rounding operations, Algorithm 2 is a heuristic algorithm. Therefore, it does not guarantee the globally optimal solution of problem P2. Specifically, if the subgradient method is adopted to obtain the optimal Lagrangian multipliers, the computation complexity of Algorithm 2 can be characterized by $O\left(MK/\left(1/\xi^{2}\right)\right)$, where the ξ is the tolerance of Lagrangian multipliers solution. Moreover, Algorithm 2 is implemented based on two kinds of information: (1) the ID of the content requested by each user, (2) the data rates that can be achieved by each user when it is associated with one specific SBS. The users should upload the content ID and the channel power channel between itself to the SBSs in a real-time manner. Besides, although Algorithm 2 is proposed based on the assumption that users’ location would be fixed during one time slot, it is still able to cope with the dynamics due to users’ movement. Specifically, for the scenario that users are with low mobility, the channel power gain used in the decision process should be the average value over the following one time slot or several time slots. Specifically, the channel power gain information can be predicted based on machine learning techniques. For the scenario that users are with high mobility, the frequency of implementing Algorithm 2 should be improved to cope with the rapid fluctuation of the wireless channel condition.

**Algorithm 2** Cache-aware User Association and Resources Allocation Algorithm**Initialize**λ,β;**repeat**     Obtain the dmk* based on Equation (Equation 14);     Update λ,β based on ellipsoid method or subgradient method;     T=T+1;**until** Convergence or T≥Tmax;Obtain amk* and bmk* based on Equations (Equation 16) and (Equation 17), respectively.


Furthermore, by combining the closed-form solution of problem P3, i.e., Equation (Equation 15), with the complementary slackness conditions, we can obtain more insights into the characteristics of the user association and resources allocation solution. Specifically, when the optimal user association and resources allocation solution is obtained, the SBS would be in one of the three states which is shown in the following:State One: The SBS has enough backhaul-bandwidth, but the available RBs are not sufficient for all the associated user, i.e.,
(18)$$\left\{\begin{array}{c} \sum_{k\in K}c_{m}^{k}d_{m}^{k*}NR_{m}^{k}<B_{m}^{BH},\\ \sum_{k\in K}d_{m}^{k*}=1. \end{array}\right.$$
which can be called as a backhaul-sufficient and RB-constrained state.Based on the complementary slackness conditions, it can be derived that
(19)$$\left\{\begin{array}{c} \beta_{m}^{*}=0,\\ \lambda_{m}^{*}\geq 0, \end{array}\right.$$
Then, the optimal solution would be
(20)$$d_{m}^{k*}=\frac{1}{\lambda_{m}^{*}},$$
which means that each user associated with SBS *m* would be allocated with the same number of RBs. This is consistent with the conclusion of paper [12], i.e., equal allocation of the frequency bandwidth is optimal for load-aware user association problem where the backhaul capacity is sufficient.State Two: The SBS is backhaul-constrained but the available RBs are sufficient, i.e.,
(21)$$\left\{\begin{array}{c} \sum_{k\in K}c_{m}^{k}d_{m}^{k*}NR_{m}^{k}=B_{m}^{BH},\\ \sum_{k\in K}d_{m}^{k*}<1, \end{array}\right.$$
which can be called as backhaul-constrained and RB-sufficient state.According to the complementary slackness conditions, the following can be derived, i.e.,
(22)$$\left\{\begin{array}{c} \beta_{m}^{*}\geq 0,\\ \lambda_{m}^{*}=0. \end{array}\right.$$
Then, the optimal solution would be
(23)$$d_{m}^{k*}=\frac{1}{\beta_{m}^{*}NR_{m}^{k}c_{m}^{k}}.$$
Based on (Equation 23), two conclusions can be deduced for the SBSs in backhaul-constrained and RB-sufficient state. Firstly, none of the contents requested by the associated users has been cached. This can be proved as follows. If the SBS is RB-sufficient, i.e.,$\sum_{m\in M}d_{m}^{k*}<1$, there would be redundant RBs that are not allocated to any of the associated users. Suppose that there was one user whose requested content was cached, the redundant RBs could be allocated to it and the overall utility of the SBS could also be improved. This would lead to $\sum_{m\in M}d_{m}^{k*}=1$, which is in contradiction with the RBs-constrained assumption. Consequently, it can be concluded that all the requests of the associated users miss the cache. Secondly, the number of RBs allocated to the associated users would be inversely proportional to their data rates of the wireless link, and the backhaul bandwidth would be allocated equally among the associated users. This can be easily derived based on Equation (Equation 23) and means that data rates of each user would be equal when the backhaul link is the bottleneck of the transmission and the wireless link is sufficient.State Three: Consider the SBS is both backhaul-constrained and RB-constrained, i.e.,
(24)$$\left\{\begin{array}{c} \sum_{k\in K}c_{m}^{k}d_{m}^{k*}NR_{m}^{k}=B_{m}^{BH},\\ \sum_{k\in K}d_{m}^{k*}=1, \end{array}\right.$$
the following can be derived, i.e.,
(25)$$\left\{\begin{array}{c} \beta_{m}^{*}\geq 0,\\ \lambda_{m}^{*}\geq 0, \end{array}\right.$$
Then the optimal solution would be in the same form with Equation (Equation 14), i.e.,
(26)$$d_{m}^{k*}=\frac{1}{\lambda_{m}^{*}+\beta_{m}^{*}NR_{m}^{k}c_{m}^{k}}.$$
Note that if there are requests hit the cache, a part of the available RBs would be equally allocated among the corresponding users. The remaining RBs would be allocated to the associated users of cache miss. However, different from state two, the fraction of the RBs allocated to the users of cache miss is not adversely proportional to their wireless data data rates, except that all the requests of the associated users miss the cache. In addition, the fraction of the overall RBS allocated to users of cache hit is larger than that of the users of cache miss.

## 5. Simulation Results

In this section, the performance of the proposed algorithms is evaluated based on simulation. The simulation is implemented by MATLAB R2018 on a laptop with four 1.9 GHz CPU cores and 16 GB of memory. Specifically, as shown in Figure 1, a UDN scenario with grid topology is considered, where 9 SBSs are deployed in a 60 m × 60 m area. The system bandwidth is 20 MHz, and is divided into 100 RBs. The transmit power of each SBS is fixed as 23 dBm. The user-SBS association is implemented in a large time scale compared to the change of the channel, and the users are assumed to be fixed without mobility during the slot of one content delivery. So only the pathloss and the shadowing are considered in the channel model. The L(d)=37+30log(d) is used as the pathloss model, where *d* is the distance between the user and the SBS. The shadowing standard deviation between SBSs and the users is 10 dB. The power density of noise is −174 dBm. The user number is fixed as 100, each with the minimum data rates requirement of 5 Mbps. The main system parameters are summarized in Table 1.

In order to reduce the simulation time, the total number of the contents in the library is set as 1000, i.e., F=1000. The probability of user *k* requesting content *f* is modeled by Zipf distribution, i.e.,
(27)$$p^{k,f}=\frac{{\left(\phi \left(k\right)\right)}^{-\gamma }}{\sum_{f\in F}f^{-\gamma }},$$
where the shape parameter γ is 0.8. In particular, in order to characterize users’ different content preferences, different rank is assumed for each user. This is enabled by the function $\phi \left(k\right)$ in Equation (Equation 27), which would return a random permutation of [1,2,⋯,F] [28]. Based on this, each user would have different request probability for the contents.

Furthermore, it has been shown in [21] that 80% of the users send video requests from less than four locations. So in the simulation there would be four SBSs for each user from which they would send the contents, and the probability of the user sending requests from one of the four SBSs is assumed to be equal.

**Baseline Algorithms** Two baseline algorithms are also simulated simultaneously to make comparison with the proposed algorithms of this paper (which would be called as PA in the following).

**Baseline 1** The local-popularity based content placement scheme, SINR-Max based user-SBS association scheme and equal resources allocation scheme are adopted as baseline Algorithm 1 (which would be called as BA 1 in the following). Specifically, the local-popularity-based content placement means that SBS *m* would cache the contents with the highest value of $p_{m}^{f}=\sum_{k\in K}{\mathrm{Pr}}_{m}^{k}p^{k,f}$ until the cache is full [29,30]. The SINR-Max user-SBS association means that each user would be associated with the SBS with the highest SINR [12]. The fair resources allocation scheme means that the wireless frequency bandwidth would be allocated equally among the associated users, and the backhaul bandwidth would be allocated equally among the associated users of cache miss.**Baseline 2** The iterative content placement and user association algorithm proposed in [16] is adopted as the baseline Algorithm 2 and would be called as BA 2 in the following. Specifically, the BA 2 can maximize the weighted sum of users’ data rates utility and the backhaul traffic reduction (The value of the weight is set as 10 [16]). During the content placement phase, the iterative content placement and user association procedures of BA 2 would be implemented. As the BA 2 is designed based on the assumption that each user has one specific location, it cannot be applied to the scenarios where the users tend to have multi-location request patterns. Thus, one of the possible locations would be chosen randomly and fixed as the user’s location during the content placement phase. As for the content delivery phase, only the user association procedure would be implemented and the backhaul bandwidth would be allocated equally among the associated users of cache miss.

**Performance Evaluation Metrics** Three performance metrics are evaluated, which are listed as follows:The users’ data rates utility, which is the sum of the logarithmic user’s data rates (i.e., the objective function of P2) and can be calculated by $U=\sum_{k\in K}\log\sum_{m\in M}a_{m}^{k}b_{m}^{k}R_{m}^{k}$.The percentage of QoS satisfied users, which is the ratio of the users whose data rates are larger than their minimum data rates requirements. It can be calculated based on $\eta =\frac{1}{K}\sum_{k\in K}\Gamma \left(\sum_{m\in M}a_{m}^{k}b_{m}^{k}R_{m}^{k}>R^{k,TH}\right)$, where the function $\Gamma \left(•\right)$ is defined as $\Gamma \left(x>y\right)=\left\{\begin{array}{c} 1,\mathrm{if}\;\;x>y,\\ 0,\mathrm{else}. \end{array}\right.$The hit ratio, which is the ratio of users of cache hit, and can be calculated based on γ=∑m∈Mcmkamk∑m∈McmkamkKK.

### 5.1. Impact of the Backhaul Bandwidth

Firstly, the performance of the algorithms is evaluated with different backhaul bandwidth constraints. Specifically, the cache capacity is fixed as 100, which means 10% of the overall contents could be cached by each SBS. The simulation results are averaged over 100 instances, and every user would send 10 content requests in each instance. During the delivery of one content, users’ locations are fixed, however, their locations may be different from one content delivery to the other.

The users’ data rates utility is shown in Figure 2. It can be seen that the utilities of all the three algorithms grow with the increase of the backhaul bandwidth. This validates that with low backhaul bandwidth, the backhaul link would be the bottleneck that limit both users’ and network’s performance. With the alleviation of the backhaul bandwidth strain, the potential of the network performance could be utilized more sufficiently, and users’ performance can be improved. Besides, the performance of the PA is larger than any one of the baseline algorithms, which verifies PA’s great advantage over the existing algorithms. Furthermore, the performance of BA 2 is better than that of the BA 1 when the backhaul bandwidth is below 20 MHz, but becomes lower than that of the BA 1 when the backhaul bandwidth is larger. This verifies that the BA 1 is more suitable to cope with the backhaul-constrained scenarios.

The performance in terms of the percentage of QoS-satisfied users is shown in Figure 3. Similarly to users’ data rates utility, the percentage of QoS satisfied users improves with the increase of the backhaul bandwidth for all the three algorithms. Moreover, it is obvious that the PA has the better performance all the time, compared with the BA 1 and BA 2. However, in contrast with the users’ data rates utility, the percentage of QoS-satisfied users of PA increases very slowly when the backhaul bandwidth is smaller than 25 Mbps. This is because PA tends to achieve a proportional-fair user association and resources allocation solution, and the data rates of most of the users’ would approach the minimum data rates simultaneously. However, if their data rates are not larger than the minimum data rates requirements, this increase would not improve the percentage of QoS-satisfied users. When the backhaul bandwidth is between 30 Mbps to 50 Mbps, the percentage of QoS-satisfied users increases sharply. This means that a large number of users exceed the minimum data rates requirements, with the continuous improvement of backhaul bandwidth. When the backhaul bandwidth is larger than 55 Mbps, the percentage of QoS-satisfied users returns to a slow increase. This is consistent with the trend of data rates utility shown in Figure 2. The stalled increase means that the backhaul bandwidth is large enough and the backhaul link is not the bottleneck of the content delivery any more.

The hit ratio performance of the three algorithms is shown in Figure 4. It can be seen that the hit ratio of both BA 1 and BA 2 does not change when the backhaul bandwidth increases. This is because the user association and content placement of the baseline algorithms have no dependency on the backhaul bandwidth constraint. As a result, the corresponding hit ratio of the BA 1 and BA 2 is not impacted by the backhaul bandwidth. In contrast with the intuition, the Figure 4 shows that the hit ratio of PA decreases when the backhaul bandwidth becomes larger. This is due to the fact that the user association and resources allocation of PA is backhaul bandwidth-aware. When the backhaul bandwidth is not large enough, the backhaul link would be the bottleneck limiting users’ performance. Without sufficient backhaul bandwidth, the users tend to be associated with the SBSs which are not close to them but have cached the requested contents to reduce the backhaul traffic. This would lead to good performance of hit ratio, but the users’ data rates utility would not be improved significantly. However, with the increase of the backhaul bandwidth, the backhaul link can deliver more traffic and the dependency of content delivery on SBSs’ cache is reduced. This means that the users would be associated with neighboring SBSs but can still get good data rates utility. Consequently, with the growth of the backhaul bandwidth, the hit ratio would decline but the data rates utility would increase. Based on this, it can be deduced that the PA has the flexibility of making full use of the (1) communication resources of both the wireless and backhaul links, and (2) the storage resources of SBSs’ cache, to improve the users’ performance.

### 5.2. The Impact of the Cache Capacity

The impact of the cache capacity on the algorithms’ performance is evaluated in this subsection. Specifically, as this paper focuses on the backhaul-constrained scenario, the backhaul bandwidth of each SBS is fixed as 5 Mbps. The simulation results are also averaged over 100 instances, and each user would request 10 contents in one instance. During the delivery of one content, users’ locations are fixed, however, their locations may be different from one content delivery to the other.

The users’ data rates utility, the percentage of QoS satisfied users, and the hit ratio for different cache capacity are shown in Figure 5, Figure 6 and Figure 7, respectively. It can be seen that the users’ data rates utilities, the percentage of QoS-satisfied users, and the hit ratio for all the three algorithms increase with the expand of the cache capacity. This is consistent with the intuition that with larger cache capacity, more contents can be cached by each SBS, and the probability of cache hit would be improved. This is critical for the backhaul-constrained UDN scenarios, which would relieve the backhaul burden and improve users’ data rates. Additionally, it can be seen that the PA outperforms the BA 1 and BA 2 in terms of users’ data rate utility and the percentage of QoS satisfied users all the time, which shows the advantage of the PA over the existing algorithms.

Interestingly, the performance of BA 2 in terms of hit ratio is better than that of the PA, except the scenario where the cache capacity is 1000 (i.e., all the contents can be cached by each SBS). However, the users’ data rates utility and percentage of QoS-satisfied users of BA 2 are always smaller than those of PA’s, which means that the high hit ratio of BA 2 brings no performance improvement from the perspective of users data rates. The performance gap between the PA and the BA 2 is caused by two facts. The first one is that BA 2 is designed based on the assumption that each user would send requests from one specific location, which can not adapt to users’ multi-location requests pattern. This verifies the necessity of considering users’ multi-location property in the content placement scheme, which is one of our main contributions in this paper. The second one is that the BA 2 has a weighted sum utility function, of which the weight is fixed and lacks the the flexibility to adapt to different network scenarios. On the other hand, the PA proposed in this paper is a backhaul bandwidth-aware and cache-aware user association and resources allocation scheme, which is adaptive to diverse network scenarios with different cache capacity and backhaul bandwidth constraints.

### 5.3. The Impact of SBSs’ and Users’ Number

In this subsection, the impact of the users’ and SBSs’ number on the network performance is evaluated. Specifically, two grid network scenarios with 9 SBSs and 16 SBSs are simulated, respectively, where all the SBSs are distributed in a 60 m × 60 m area. Besides, the users’ number is varied from 50 to 100. The cache capacity is fixed as 100. The backhaul bandwidth of each SBS is set as 5 Mbps. The simulation results are averaged over 100 instances, and every user would send 10 content requests in each instance. During the delivery of one content, users’ locations are fixed, however, their locations may be different from one content delivery to the other.

The average utility of user’s data rates and the percentage of QoS-satisfied users are shown in Figure 8 and Figure 9, respectively. It is obvious that with the increase of users’ number, both the metrics decrease for all the algorithms no matter how many SBSs there are. This is intuitive because with limited cache capacity and backhaul bandwidth, the more users, the less backhaul and wireless bandwidth could be shared by each user. As a result, the average user’s date rates, as well as the percentage of QoS-satisfied users would be smaller. Besides, when the number of SBSs is increased from 9 to 16, the average utility of each user and the percentage of QoS-satisfied users are also improved. This is because when there are more SBSs, the available cache storage resources, as well as the wireless and backhaul bandwidth would increase. As a result, the users can benefit from this and the average data rates would be improved. Moreover, Figure 8 and Figure 9 show that the PA achieves the best performance in terms of both average utility and the percentage QoS-satisfied users no matter there are 9 SBSs or 16 SBSs in the network. This reveals the advantage of the PA in utilizing the limited cache storage and communication resources to improve each user’s utility.

The hit ratio performance for different number of users and SBSs is shown in Figure 10. It can be seen that for all the three algorithms, the hit ratio performance decreases slightly with the increase of the number of users. This is because that the cache capacity of each SBS is fixed and limited but the users’ preferences are diverse. It would be more difficult to utilize the limited cache capacity to store the proper contents when the users number becomes larger. Consequently, the hit ratio performance would deteriorate. Furthermore, note that when the number of SBSs increases from 9 to 16, the corresponding hit ratio performance of all the three algorithms increases. This is intuitive because more SBSs would lead to more storage resources for the network. Hence, there would be higher probability for each user to be served by SBSs’ cache. Besides, note that the BA 2 achieves the highest hit ratio among all the algorithms, no matter how many users and SBSs there are. But combined with the Figure 8 and Figure 9, it can be derived that the highest hit ratio performance of BA 2 is at the cost of each user’s data rates degradation.

## 6. Conclusions

In this paper, the proactive content caching and content delivery problem is investigated for the cache-enabled UDN with constrained-backhaul link. Specifically, the content placement problem is formulated for the proactive caching phase, where the multi-location request pattern of mobile users is considered. Meanwhile, the user-SBS association and resources allocation problem is formulated for the content delivery phase, where the wireless and backhaul link resources constraints are considered. Due to the non-convex nature of the problems, the two problems are transformed and converted into tractable forms, and then the content placement algorithm, user-SBS association and resources allocation scheme are proposed, respectively. Finally, the simulation results show that the proposed algorithms have obvious performance advantage in terms of network utility, the hit ratio, and QoS provisioning.

## Figures and Tables

**Figure 1 sensors-19-02449-f001:**
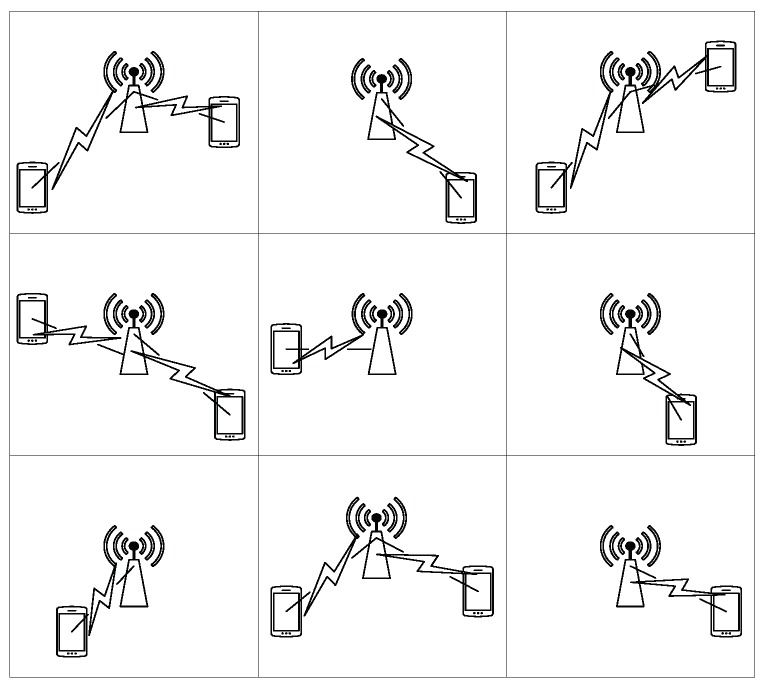
Network scenario used in simulations.

**Figure 2 sensors-19-02449-f002:**
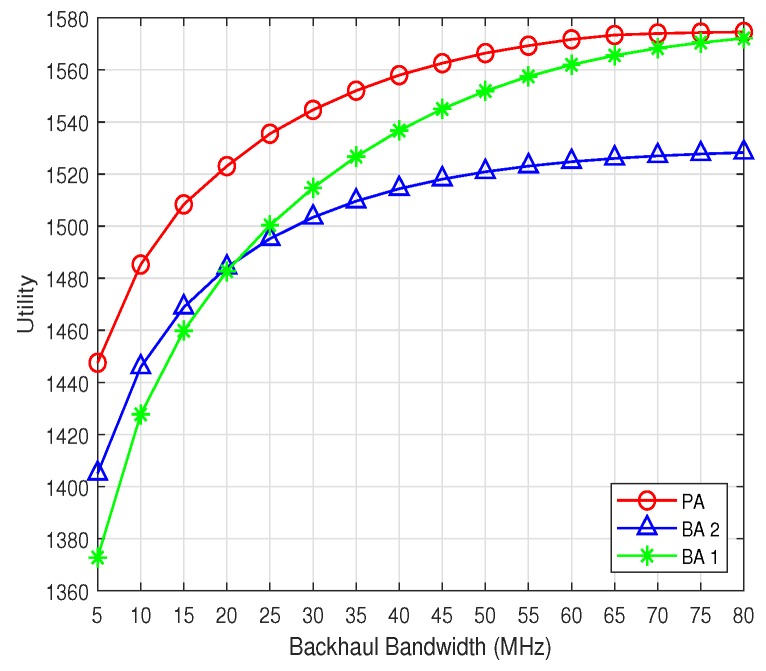
The network utility.

**Figure 3 sensors-19-02449-f003:**
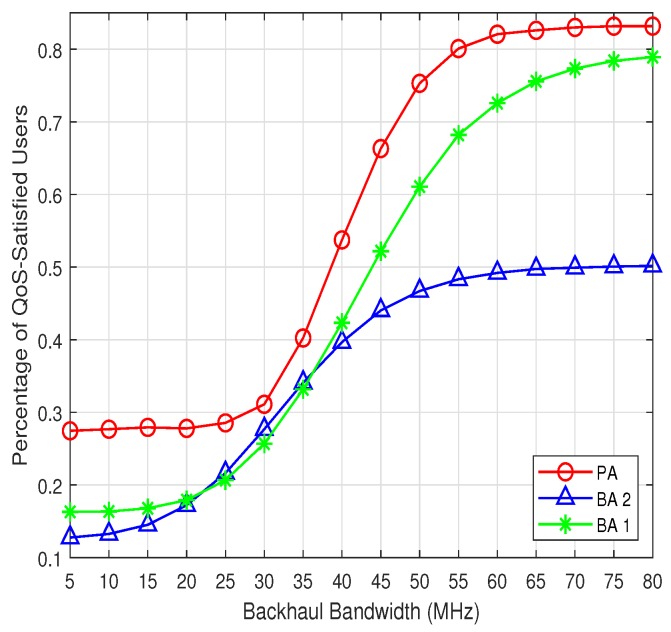
The percentage of quality of service (QoS)-satisfied users.

**Figure 4 sensors-19-02449-f004:**
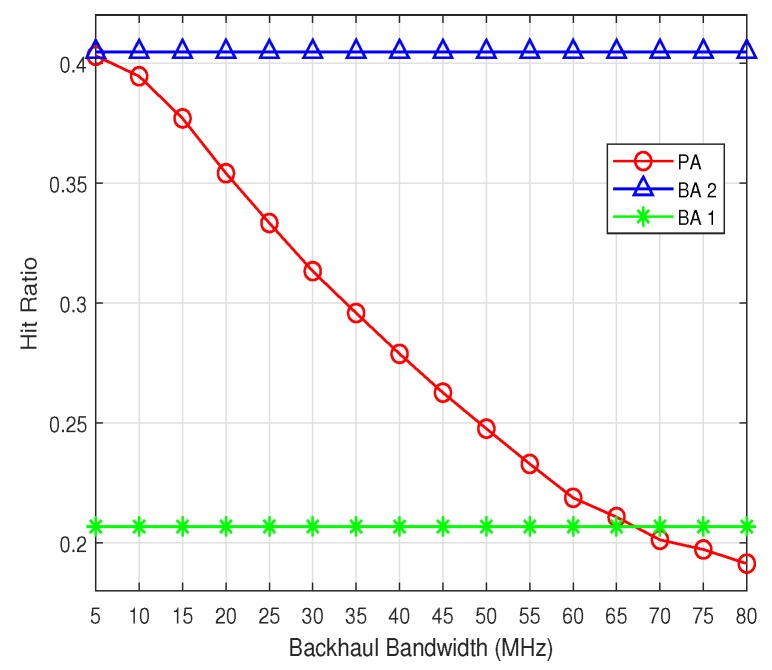
The hit ratio.

**Figure 5 sensors-19-02449-f005:**
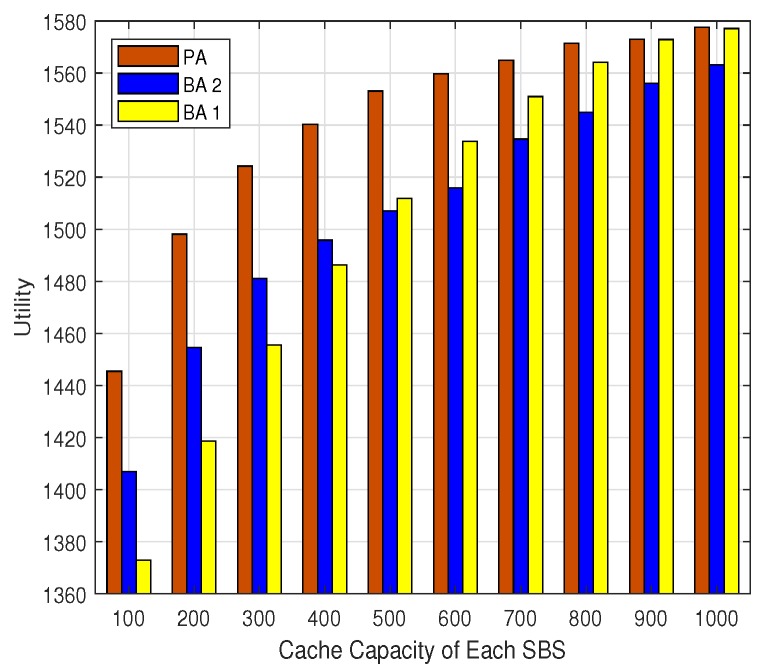
The network utility.

**Figure 6 sensors-19-02449-f006:**
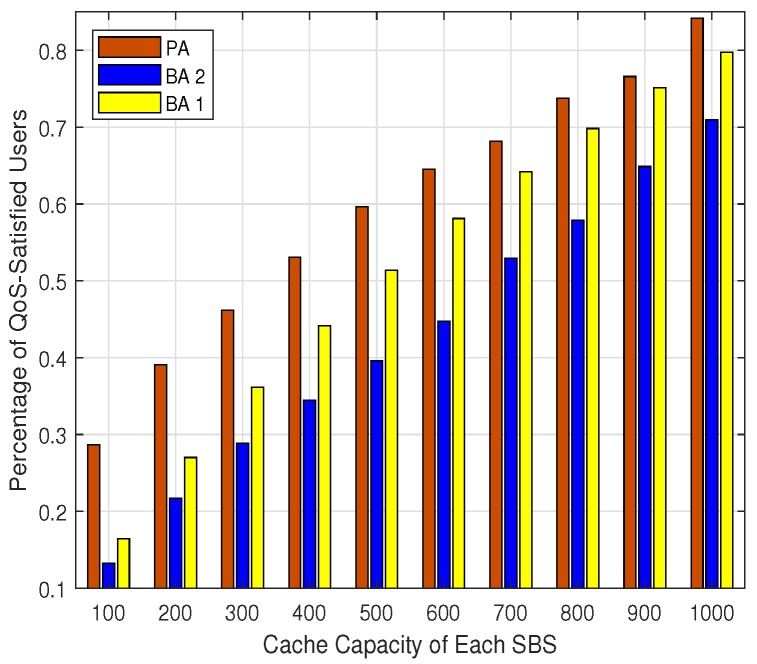
The percentage of QoS-satisfied users.

**Figure 7 sensors-19-02449-f007:**
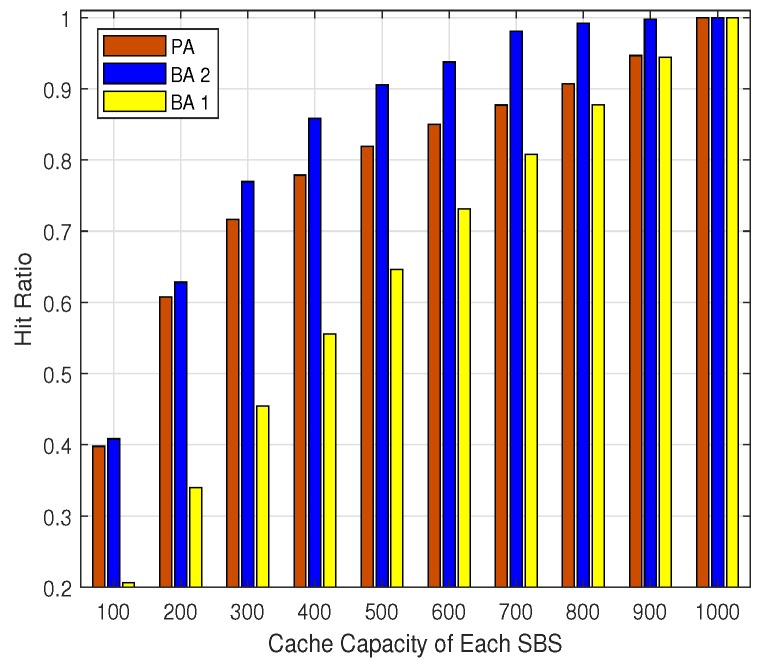
The hit ratio.

**Figure 8 sensors-19-02449-f008:**
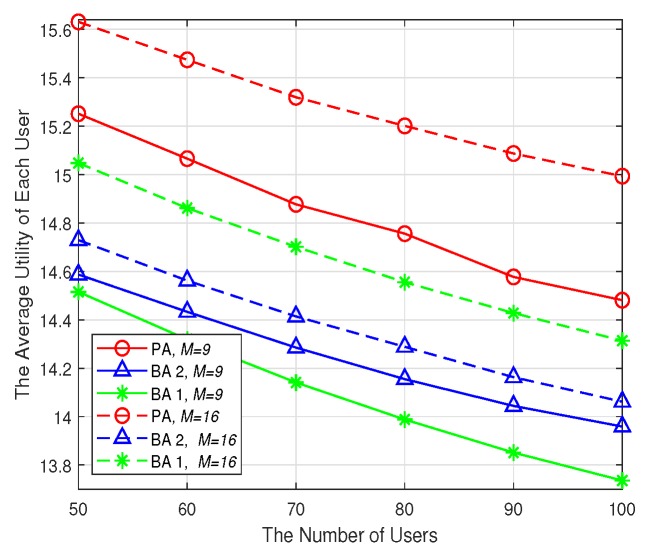
The network utility.

**Figure 9 sensors-19-02449-f009:**
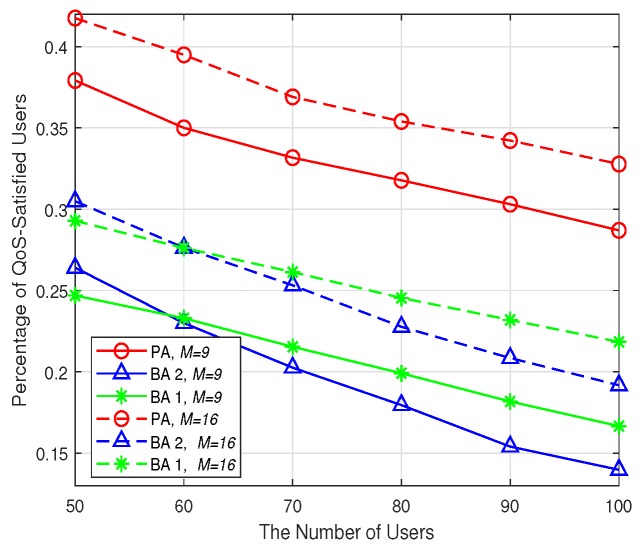
The percentage of QoS-satisfied users.

**Figure 10 sensors-19-02449-f010:**
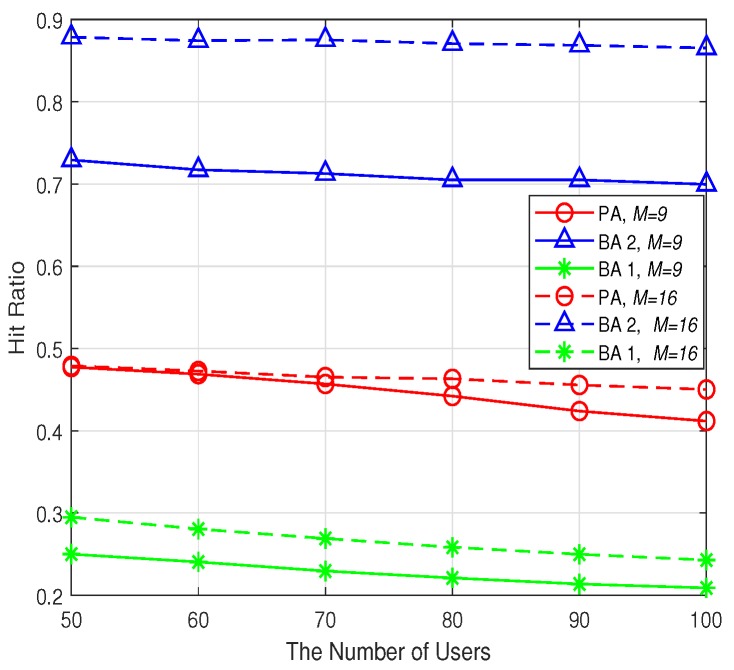
The hit ratio.

**Table 1 sensors-19-02449-t001:** Simulation Parameters.

Parameter	Value
Number of SBSs	9
Number of users	100
System bandwidth	20 MHz
The number of RBs	100
The number of contents	1000
SBS’s transmit power	23 dBm
The power density of noise	−174 dBm

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
