# Peer review of "Multi-Location-Aware Joint Optimization of Content Caching and Delivery for Backhaul-Constrained UDN"

_sensors, 2019, doi:10.3390/s19112449_

Round 1
Reviewer 1 Report
- In this paper, the authors investigate the optimization of content placement and content delivery schemes in cache-enabled UDN with constrained-backhaul.
- This is an interesting topic, worthy to discuss.
- This is a well-written paper. The contents, problems, and solution were provided very clearly. The results and its explanations were also presented in detail.
- However, this paper still has remained some language flaws. In the detail, the “hence”, “note that”, “beside”, etc. were used many times (more than 15 times) in the paper. The authors can use another synonym words to replace.
Author Response
The response is list in the word file.

Reviewer 2 Report
This paper shows the efficient content caching management scheme in small cell networks based on mobile edge environments.
It needs to clarify the following points.
Dynamics due to user movements
- When considering the frequency of node movement, the association of user-BS is expected to be dynamically changed. So, it needs to show how much the proposed scheme is adapted against such dynamic condition.
Density of SMS and users
- It is well known that the cache placement method is affected by the user density. Moreover, duplicate cache or cache hit ratio are influenced by the number of SBS. So, this paper needs to show the effect or tradeoff for such situation
Duplicate caching
- In SBS with resource-constraint, fixed periodic information exchange for the cached content cause the cache duplicates.
Author Response
Authors' response to Reviewer of “Multi-Location-Aware Joint Optimization of Content Caching and Delivery for Backhaul-constrained UDN” (sensors-493629)
Authors: Many thanks for your precious time and efforts expended in an attempt to improve our paper. Your insightful advice is very much appreciated. We attempted to address all your concerns, which doubtlessly improved the paper and our hope is that our efforts will meet your approval. Our detailed reflections are listed below point by point.
1.Reviewer: This paper shows the efficient content caching management scheme in small cell networks based on mobile edge environments. It needs to clarify the following points.
- Dynamics due to user movements: When considering the frequency of node movement, the association of user-BS is expected to be dynamically changed. So, it needs to show how much the proposed scheme is adapted against such dynamic condition.
Authors: Thank you very much for seeking more clarification on the issue of the proposed scheme’s adaptation against dynamic condition.
Indeed, the position of the users are not fixed and users would move frequently in the wireless networks. As a result, their association with BSs may need adjustment frequently.
In our paper, the scenario is considered that each user would send content request in a slot-by-slot manner. In one specific slot, the user’s location is assumed to be fixed.
Although the proposed user-SBS association algorithm is based on the assumption that user’s location would be fixed during one time slot, the proposed user-SBS algorithm can cope with the dynamics due to user movement with minor modifications. Specifically,
1) For the scenario that users are with low mobility, which means users move at low velocity when they watch the video, the user-SBS association does not need frequent adjustment. The channel power gain used in the decision process should reflect the overall channel condition in the following time. Hence, the value of the channel power gain should be the average value over the following one time slot or several time slots. Specifically, this information can be predicted based on machine learning techniques. As a result, the user-SBS association solution obtained based on the proposed algorithm could cope with the dynamics due to users’ low-velocity movement.
2) For the scenario that users are with high mobility, which means users move at high velocity when they watch the video, the wireless channel condition may fluctuate rapidly and significantly. In order to cope with this dynamic, the frequency of implementing user-SBS association algorithm need be improved. In particular, the real-time channel information should be uploaded frequently for calculate the user-SBS association solutions.
We clarify this issue by adding the following sentences in the Subsection 4.2 of the revised manuscript.
Besides, although the Algorithm 2 is proposed based on the assumption that users' location would be fixed during one time slot, it is still able to cope with the dynamics due to users' movement. Specifically, for the scenario that users are with low mobility, the channel power gain used in the decision process should be the average value over the following one time slot or several time slots. Specifically, the channel power gain information can be predicted based on machine learning techniques. For the scenario that users are with high mobility, the frequency of implementing Algorithm 2 should be improved to cope with the rapid fluctuation of the wireless channel condition.
2. Reviewer: Density of SMS and users: It is well known that the cache placement method is affected by the user density. Moreover, duplicate cache or cache hit ratio are influenced by the number of SBS. So, this paper needs to show the effect or tradeoff for such situation.
Authors: Thank you very much for your valuable advices. We are sorry for that we did not analyze the impact of user density and the number of SBSs on the network performance.
Heeding your advices, we do the following simulation to investigate the impact of user density and SBSs number. Specifically, two grid network scenarios with 9 SBSs and 16 SBSs are simulated, respectively, where all the SBSs are distributed in a 60m × 60m area. Besides, the users’ number is varied from 50 to 100. The cache capacity is fixed as 100. The backhaul bandwidth of each SBS is set as 5Mbps. The simulation results are averaged over 100 instances, and every user would send 10 content requests in each instance. During the delivery of one content, users’ locations are fixed, however, their locations may be different from one content delivery to the other.
The average utility of each user, the percentage of QoS-satisfied users, and the hit ratio are shown in the figures below. Specifically, the main phenomena are summarized as follows:
Fig.1 The average utility of each user Fig.2 The percentage of QoS-satisfied users Fig.3 The hit ratio
1) With the increase of users’ number, both the average utility of each user and the percentage of QoS-satisfied users decrease for all the algorithms no matter how many SBSs there are. This is intuitive because with limited cache capacity and backhaul bandwidth, more users in the network means less backhaul and wireless bandwidth could be shared by each user. As a result, the average user’s date rates, as well as the percentage of QoS-satisfied users would decrease.
2) For all the three algorithms, when the number of SBSs is increased from 9 to 16, both the average utility of each user and the percentage of QoS-satisfied users are improved. This is because when there are more SBSs, the available cache storage resources, as well as the wireless and backhaul bandwidth would increase. All the users can benefit from this and the average data rates would be improved.
3) The PA achieves the best performance in terms of both the average utility of each user and the percentage QoS-satisfied users, no matter there are 9 SBSs or 16 SBSs in the network. This reveals that the advantage of the PA in utilizing the limited cache storage resources and communication resources to improve each user's utility.
4) For all the three algorithms, the hit ratio performance decreases slightly with the growth of the number of users. This is because that the cache capacity of each SBS is fixed and limited, but the users' preferences are diverse. It would be more difficult to utilize the limited storage capacity to cache the proper contents when the user number becomes larger. Consequently, the hit ratio performance would deteriorate with the growth of the user number.
5) When the number of SBSs increases from 9 to 16, the corresponding hit ratio performance of all the three algorithms increases. This is because more SBSs would lead to more storage capacity for the network. Hence, there would be higher probability for each user to be served by SBSs' cache.
6) Besides, note that the BA 2 achieves the highest hit ratio among all the algorithms, no matter how many users and SBSs there are. But combined with the Fig. 1 and Fig. 2, it can be derived that the highest hit ratio performance of BA 2 is at the cost of each user's data rates degradation.
We updated the manuscript by adding the figures above and the following paragraphs in the revised manuscript as the Subsection 5.3.
5.3 The impact of SBSs' and users' number
In this subsection, the impact of the users' and SBSs' number on the network performance is evaluated. Specifically, two grid network scenarios with 9 SBSs and 16 SBSs are simulated, respectively, where all the SBSs are distributed in a 60m×60m area. Besides, the users’ number is varied from 50 to 100. The cache capacity is fixed as 100. The backhaul bandwidth of each SBS is set as 5Mbps. The simulation results are averaged over 100 instances, and every user would send 10 content requests in each instance. During the delivery of one content, users’ locations are fixed, however, their locations may be different from one content delivery to the other.
The average utility of user's data rates and the percentage of QoS-satisfied users are shown in Fig.8 and Fig.9, respectively. It is obvious that with the increase of users' number, both the metrics decrease for all the algorithms no matter how many SBSs there are. This is intuitive because with limited cache capacity and backhaul bandwidth, the more users, the less backhaul and wireless bandwidth could be shared by each user. As a result, the average user's date rates, as well as the percentage of QoS-satisfied users would be smaller. Besides, when the number of SBSs is increased from 9 to 16, the average utility of each user and the percentage of QoS-satisfied users are also improved. This is because when there are more SBSs, the available cache storage resources, as well as the wireless and backhaul bandwidth would increase. As a result, the users can benefit from this and the average data rates would be improved. Moreover, Fig.8 and Fig.9 show that the PA achieves the best performance in terms of both average utility and the percentage QoS-satisfied users no matter there are 9 SBSs or 16 SBSs in the network. This reveals the advantage of the PA in utilizing the limited cache storage and communication resources to improve each user's utility.
The hit ratio performance for different number of users and SBSs is shown in Fig. 10. It can be seen that for all the three algorithms, the hit ratio performance decreases slightly with the increase of the number of users. This is because that the cache capacity of each SBS is fixed and limited but the users' preferences are diverse. It would be more difficult to utilize the limited cache capacity to store the proper contents when the users number becomes larger. Consequently, the hit ratio performance would deteriorate. Furthermore, note that when the number of SBSs increases from 9 to 16, the corresponding hit ratio performance of all the three algorithms increases. This is intuitive because more SBSs would lead to more storage resources for the network. Hence, there would be higher probability for each user to be served by SBSs' cache. Besides, note that the BA 2 achieves the highest hit ratio among all the algorithms, no matter how many users and SBSs there are. But combined with the Fig. 8 and Fig. 9, it can be derived that the highest hit ratio performance of BA 2 is at the cost of each user's data rates degradation.
3. Reviewer: Duplicate caching: In SBS with resource-constraint, fixed periodic information exchange for the cached content cause the cache duplicates.
Authors: Thank you very much for seeking more clarification on the duplicate caching issues.
The content placement algorithm proposed in the manuscript is a centralized algorithm, where all the procedures are implemented in a central entity, e.g., the remote server of the content provider. Specifically, the following information is required to calculate the content placement solution: 1) where the user would send the content request, i.e., , 2) which content the user would request for, i.e., , and 3) the data rates of the wireless link between the SBS and the user, i.e., . Note that all of the information could be estimated based on the history records of users’ requests. For each content request log, the location from which the user send the request, the content ID, and the data rates of the transmission link could be used to estimate the ,and.
Indeed, the information of 1) the location from which the user sends the request, as well as the 2) the content ID, could be obtained by the content provider server in a real time manner. The reason is that when the user sends content request, the request would be sent to the server of the content provider firstly, which is similar with the traditional CDN. Then, the content provider would decide to response the request by the remote server or by the cache of the SBS the user associated with. As a result, the content provider would know all the information of the request, including user’s ID, the requested content ID, the time and the location of the user sending the request.
As for the data rates of the wireless link between the SBS and the user, it should be uploaded by the SBS to the central entity, i.e., the content provider server. With the purpose of reducing the signaling exchange, we proposed a periodic information exchange mechanism in the original manuscript, which is shown as follows:
But this information exchange can be implemented periodically with a large time scale, rather than in a real-time manner.
However, the effectiveness of the data rates information could not be guaranteed. Without the timely information, the content provider would fail to track the fluctuation of transmission link characteristics. In order to make sure that the information is up-to-date, the frequency for each SBS to upload the average data rates information should be large enough, e.g., in a slot by slot manner. Then, the information based on which the content provider made the content placement decision would be comprehensive and up-to-date. Consequently, the duplicate cache problem can be solved to some extent.
We clarify this issue by modifying the sentences in the last paragraph of Subsection 4.1 in the revised manuscript as follows:
Besides, the Algorithm 1 is a centralized algorithm, where all the procedures are implemented in a central entity, e.g., the remote server of the content provider. Specifically, the information of ,and is required to calculate the solution of P1.Note that similar with the traditional CDN, when the user sends request for certain content, the request would be sent to the server of the content provider firstly. Then, the content provider would decide to response the request by the remote server or by the cache of SBS the user associated with. Hence, the content provider would know all the information of the request, including user’s ID, the requested content ID, the time and the location of the user sending the request. Furthermore, all this information is known by the content provider in a real-time manner. Moreover, as long as the average data rates of the wireless link could be uploaded in a real-time manner by SBSs, the information based on which the content provider made the content placement decision, would be comprehensive and up-to-date. Consequently, the Algorithm 1 could avoid the duplicate cache problem to some extent.
Again, thank you very much for your insightful suggestions to our paper.
Sincerely: The Authors

Reviewer 3 Report
Some English errors:
page 3, line 88:
- change "investigate optimizing the caching strategy .." to ".. investigate the optimization of the caching …"
page 7, line 227:
- change "which mean.." to "which means.."
page 8, line 252:
- change "would be reduced to be the following.." to "would be reduced to the following.."
page 11, line 293:
- change "Furthermore, combined.." to "Furthermore, by combining …"
page 3, line 119: rephrase "it remains to be a challenge problem .."
Pages 16-17
I think Figures 5,6,7 should be aligned somehow (not simply shown one below the other, because this affects readability of the results)
Error in all the figures: change "BA2" to "BA 2" (because this is how is referred in the text).
page 3: I would change the title of Section 1.2 to "Contribution and organization of the paper" or "Our contribution and paper organization"
Error in References: need to be to reviewed/corrected.
You need to have uniform style for the Reference, e.g. Refernce 17 appears with small letters in the title of the paper while the others start with big letter for each word in the title.
References 2 and 3 are the same.
Other:
Page 1, line 29: what MIMO and mmWave abbrviations stand for ?
Page 1, line 19: put QoS abbreviation somewhere else in the paper (if needed)
Page 5, line 178: Reference 28 is an unpublished work (is an abstract) so I would remove this citation.
Page 11, line 289: change : "information:1).." to "information: 1).. " (put a space after : )
Author Response
Authors' response to Reviewer of “Multi-location-Aware Joint Optimization of Content Caching and Delivery for Backhaul-constrained UDN” (sensors-493629)
Authors: Many thanks for your precious time and efforts expended in an attempt to improve our paper. Your insightful advice is very much appreciated. We attempted to address all your concerns, which doubtlessly improved the paper and our hope is that our efforts will meet your approval. Our detailed reflections are listed below point by point.
1.Reviewer: Some English errors:
page 3, line 88: change “investigate optimizing the caching strategy ...” to “investigate the optimization of the caching …”
Page 7, line 227: change “which mean. ” to “which means.. ”
page 8, line 252: change “would be reduced to be the following.. ” to “would be reduced to the following.. ”
page 11, line 293: change “Furthermore, combined. ” to “Furthermore, by combining ”
page 3, line 119: rephrase “it remains to be a challenge problem. ”
Authors: Thank you very much for your valuable suggestions.
We are sorry for our poor English use in the manuscript. Following your advices, we have revised all the errors. Specifically,
Ø “Considering the interplay of caching and user-BS association, a few papers investigate optimizing the caching strategy and user association schemes jointly” is modified as
Considering the interplay of caching and user-BS association, a few papers investigate the optimization of the caching strategy and user association schemes jointly.
Ø “The constraints C6 and C7 are the unique association limit, which mean each user must associate with only one SBS.” is modified as
The constraints C6 and C7 are the unique association limit, which means that each user must associate with only one SBS.
Ø “Fixing the user-SBS association probability variables , problem P1 would be reduced to be the following form” is modified as
Fixing the user-SBS association probability variables , problem P1 would be reduced to the following form
Ø “Furthermore, combined the closed-form solution of problem P3, i.e., equation (15), with the complementary slackness conditions, we can obtain more insight into the characteristics of the user association and resources allocation solution” is modified as
Furthermore, by combining the closed-form solution of problem P3, i.e., equation (15), with the complementary slackness conditions, we can obtain more insights into the characteristics of the user association and resources allocation solution
Ø “Hence, it remains to be a challenge problem that how to optimize the content caching and delivery schemes by considering the user-association, users’ location uncertainty, as well as the wireless and backhaul resources constraints jointly.” is modified as
Hence, it is still a challenging problem that how to optimize the content caching and delivery schemes by considering the user-association, users’ location uncertainty, as well as the wireless and backhaul resources constraints jointly.
2. Reviewer: Pages 16-17
I think Figures 5,6,7 should be aligned somehow (not simply shown one below the other, because this affects readability of the results)
Authors: Thank you very much for your suggestion.
In the revised manuscript, we have reset the locations of the Figures 5,6,7, as well as Figures 2,3,4, and made them be aligned, respectively.
3. Reviewer: Error in all the figures: change “BA2” to “BA 2” (because this is how is referred in the text).
Authors: Many thanks for your scrutiny. We have modified “BA2” to “BA 2” in the revised manuscript.
4. Reviewer: page 3: I would change the title of Section 1.2 to “Contribution and organization of the paper” or “Our contribution and paper organization”
Authors: Thank you very much for your suggestion. We have modified the title of Section 1.2 to “Contribution and organization of the paper” in the revised paper.
5. Reviewer: Error in References: need to be to reviewed/corrected.
You need to have uniform style for the Reference, e.g. Reference 17 appears with small letters in the title of the paper while the others start with big letter for each word in the title.
References 2 and 3 are the same.
Authors: Thank you very much for your valuable advices.
We have revised the Reference part to make them in a uniform style, where each word in the title would start with big letter. Besides, the reference 3 have been removed in the revised manuscript.
6. Reviewer: Page 1, line 29: what MIMO and mmWave abbreviations stand for ?
Page 1, line 19: put QoS abbreviation somewhere else in the paper (if needed).
Authors: Thank you very much for your advices on the abbreviation issues.
MIMO stands for the Multiple-Input Multiple-Output, and mmWave stands for the millimeter wave. We have modified the related sentences in the revised paper as follows:
Furthermore, with the advance of wireless transmission technologies such as massive Multiple-Input Multiple-Output (MIMO) and millimeter wave (mmWave) communication, the backhaul link instead of the wireless link, is more likely to become the bottleneck of the data transmission link
Following your advice, we put the QoS abbreviation in the Subsection 3.1 in the revised paper, i.e.,
The in constraint C5 denotes each user's minimum data rates requirement, and the constraint C5 means that the sum of the expected RB cost to satisfy all the associated users' quality of service (QoS) requirements should not be larger than the overall available RBs.
7. Reviewer: Page 5, line 178: Reference 28 is an unpublished work (is an abstract) so I would remove this citation.
Authors: Thank you very much for your advice. We have removed it in the manuscript.
8. Reviewer: Page 11, line 289: change : “information:1)..” to “information: 1).. ” (put a space after : ).
Authors: Thank you very much for your advice. We have added a space after “information:” in the manuscript, i.e.,
Moreover, the Algorithm 2 is implemented based on two kinds of information: 1) the ID of the content requested by each user, 2) the data rates that can be achieved by each user when it is associated with one specific SBS. The users should upload the content ID and the channel power channel between itself to the SBSs in a real-time manner.
Again, thank you very much for your insightful suggestions to our paper.
Sincerely: The Authors

Round 2
Reviewer 2 Report
This revised manuscript shows many modifications and added descriptions according to review comments. Finally, some english corrections are required.